# Asleep at the Wheel: Benchmarking the Inattention of Vision Language Models to Clinical Sleep Signals

## Abstract

Clinical sleep staging, a cornerstone for diagnosing and managing sleep disorders, presents a complex multimodal challenge for artificial intelligence solutions. While Vision Language Models (VLMs) have achieved unprecedented success in general purpose multimodal understanding, their capacity to interpret long-duration, noisy physiological signals like Polysomnography (PSG) remains largely unexplored. To systematically probe this frontier, we introduce the first comprehensive benchmark, **SleepVLM-Bench**, tailored for evaluating VLM in clinical sleep staging, where we meticulously reformulate single-night multichannel PSG into VLM-consumable modalities: time and frequency images, clinician-derived textual features, and raw/discrete feature sequences. Our rigorous evaluation pits leading VLMs, including the strong multimodal model GPT-4o, against strong CNN, RNN, and Transformer baselines across three diverse, real-world cohorts (DCSM, SHHS, ISRUC) using standardized subject-wise and cross-cohort evaluations. Our central and surprising finding reveals that **state-of-the-art VLMs consistently fail** to outperform established end-to-end deep learning baselines in this critical task, and their performance remains insufficient for independent, clinical-grade deployment. Although incorporating expert feature prompting offers improved interpretability and modest robustness in low-data regimes, it does not bridge this significant performance gap. These results provide a critical, reproducible reference point, clarifying the current limitations of VLMs in handling complex, long-duration physiological data and highlighting the urgent need for future research into specialized multimodal fusion architectures, the integration of physiological priors, and efficient model designs tailored to lay the groundwork for truly robust and clinically deployable solutions with VLMs. Our dataset will be released upon acceptance.

## 1 Introduction

Clinical sleep staging, the process of classifying sleep into distinct phases (W, N1, N2, N3, REM), is a cornerstone of sleep medicine Itani et al. (2017); Finan et al. (2016). Beyond its clinical utility, it represents a quintessential challenge for artificial intelligence solutions, demanding robust long sequence understanding, multimodal signal fusion, and fine-grained temporal reasoning from noisy polysomnography (PSG) data Basha et al. (2020). For the past decade, end-to-end deep learning models, from CNNs to Transformers, have demonstrated impressive accuracy Chambon et al. (2018). Yet, they remain plagued by persistent issues: poor out-of-distribution generalization across different clinical centers and devices, an insatiable appetite for large, expert annotated datasets, and a "black-box" nature that hinders clinical trust and interpretability. These fundamental limitations have created a critical gap between research performance and real-world clinical deployment Phan et al. (2019).

The recent ascent of Large Language and Vision Language Models (LLMs/VLMs), exemplified by systems like GPT-4o Hurst et al. (2024), offers a potential paradigm shift. Their unprecedented success in general purpose reasoning, in-context learning, and multimodal understanding has fueled a powerful hypothesis Brown et al. (2020): *by reformulating complex physiological signals into modalities that VLMs excel at, such as images and natural language, we can overcome the core*

*weaknesses of previous specialized models Liu et al. (2024b)*. The promise is twofold: first, leveraging the vast world knowledge embedded in these foundation models could enhance robustness and reduce data dependency; second, by using structured textual prompts that mimic clinical heuristics, we could achieve a new level of model interpretability Singhal et al. (2023). Testing this hypothesis is not just an academic exercise; it is a critical step in determining the future role of foundation models in high-stakes medical domains Moor et al. (2023); Thirunavukarasu et al. (2023); Qi et al. (2023); Shen et al. (2023).

However, to date, **this hypothesis has remained largely untested due to the absence of a rigorous and standardized evaluation framework** Budler et al. (2025). To bridge this gap, we construct **the first systematic benchmark designed to probe the capabilities and limitations of VLMs on clinical sleep staging**. We meticulously process overnight, multi-channel PSG data into three distinct, VLM-consumable representations, each designed to test a different facet of the model's ability:

- 1) Time-Frequency Images: Visual spectrograms that challenge the model's visuo-linguistic faculties to interpret physiological patterns.
- 2) Clinician-Derived Textual Features: Natural language descriptions of EEG, EOG, and EMG events that simulate an expert's reasoning process and test the model's capacity for knowledge integration.
- 3) Raw/Discrete Feature Sequences: Direct numerical representations that assess the model's fundamental prowess in long-context temporal modeling.

Our evaluation spans three public datasets (DCSM, SHHS, ISRUC) with standardized subject-wise splits and challenging cross-cohort generalization tests, all benchmarked against strong, fairly-tuned deep learning baselines.

Our comprehensive evaluation yields a surprising and critical finding: **contrary to widespread expectations, mainstream VLMs, including strong VLMs such as GPT-4o, consistently fail to outperform specialized end-to-end baselines in fully supervised and cross-data set settings**. Their performance is fundamentally insufficient for independent, clinical-grade deployment. While we find that expert-feature prompting enhances interpretability and provides modest gains in low data scenarios, it does not close the significant performance gap. This reproducible negative result is not an endpoint but a crucial signpost for the community. It delineates a clear boundary for the current generation of VLMs, suggesting that their general purpose reasoning abilities do not seamlessly transfer to the intricate, noisy, and long duration patterns of physiological time-series data.

This work establishes a reliable reference point for future research and makes the following contributions:

1. We introduce **SleepVLM-Bench**, the first VLM-centric benchmark for clinical sleep staging, complete with a unified preprocessing pipeline, standardized data splits, and evaluation metrics, enabling fair and reproducible research.

2. We systematically investigate three distinct input modalities for adapting PSG to VLMs, clarifying the trade-offs between visual representation, textual reasoning, and raw sequence modeling.

3. We deliver a crucial and reproducible negative finding: **state-of-the-art VLMs are not yet ready for clinical-grade sleep staging**. This clarifies the limitations of current models and directs future work towards novel multimodal fusion architectures, the integration of structured physiological priors, and efficient long sequence models tailored for clinical reality.

## 2 RELATED WORKS

### 2.1 END TO END DEEP LEARNING FOR SLEEP STAGING

Convolutional, recurrent, and attention based models now dominate automated sleep staging. Early raw EEG approaches such as DeepSleepNet and its efficient successor TinySleepNet demonstrated

that compact CNN based architectures can learn discriminative filters directly from single channel inputs, with code widely used in practice Supratak et al. (2017); Supratak & Guo (2020); Zhang et al. (2025). Sequence models like SeqSleepNet reframed staging as sequence to sequence learning with hierarchical RNNs and attention, improving macro F1 and Cohen's kappa on public datasets Eldele et al. (2021). Fully convolutional designs such as U-Time achieved robust performance across multiple cohorts with a single configuration, simplifying deployment Perslev et al. (2019). Transformer based methods further advanced long range temporal modeling and offered pathways to interpretability, for example SleepTransformer and more recent flexible channel designs like FlexSleepTransformer Guo et al. (2024) that train across datasets with different montages Phan et al. (2022). Newer work explores vision transformer ideas for explainability in staging Lee et al. (2025). Despite strong within dataset results, cross center generalization remains difficult due to montage changes, sampling rate differences, and noise variability.

### 2.2 LANGUAGE MODELS AND FOUNDATION MODELS IN MEDICINE

Large language models have advanced medical text tasks such as report generation and question answering, for example Med PaLM and Med PaLM 2 on MultiMedQA, and vision language assistants adapted to biomedical figures such as LLaVA Med Singhal et al. (2023; 2022); Li et al. (2023). For time series, recent studies align or reprogram LLMs to operate on non linguistic sequences, including LLM4TS and Time LLM, yet these works focus mainly on forecasting rather than long duration clinical biosignals with stage labels Gruver et al. (2023); Jin et al. (2023). Early explorations connecting LLMs with sleep related assessments suggest potential for textual counseling while noting that detecting stages from EEG requires domain specific modeling Corda et al. (2024); Zhang et al. (2025). Overall, rigorous evaluations on long duration recordings PSG with clinical remain limited.

## 3 DATASET

### 3.1 OVERVIEW

**Curation by sleep physicians.** We evaluate three cohorts that reflect diverse centers, montages, and demographics. All recordings were scored by certified sleep physicians under AASM rules Deng S et al. (2019) and were mapped to a unified five stage taxonomy with W, N1, N2, N3, and REM. Our goal is to assess methods under realistic clinical variability while preventing information leakage. To ensure label quality and visual clarity, a panel of board certified sleep physicians screened the recordings and selected high quality epochs that exhibit canonical morphology for each stage. Nights with extensive artifacts, missing channels, or unresolved disagreements were excluded. The curated set serves as a high quality reference for evaluating models that rely on visual or textual representations.

| Dataset | Subjects | Sampling Rate | EEG Channel | EOG Channel | EMG Channel |
|---------|----------|---------------|-------------|-------------|-------------|
| DCSM | 64 | 256 | F4–M1 | E1–M2 | CHIN |
| SHHS | 15 | 125 | EEG | EOG(L) | EMG |
| ISRUC | 12 | 200 | C4–M1 | E1–M2 | CHIN |

Table 1: Overview of the processed datasets.

**SHHS** Zhang et al. (2018) The Sleep Heart Health Study is a large multi center cohort that examines links between sleep disordered breathing and cardiovascular outcomes such as coronary heart disease, stroke, all cause mortality, and hypertension. Participants were drawn from community studies including the Atherosclerosis Risk in Communities study, the Cardiovascular Health Study, the Framingham Heart Study, the New York and Cornell Hypertension Study, the Strong Heart Study, and the Tucson Epidemiologic Study. Native sampling is one hundred and twenty five hertz for EEG and chin EMG and fifty hertz for EOG referenced to PG1. ECG is sampled at one hundred and twenty five or two hundred and fifty hertz depending on the visit. For our curated subset used in modeling we retain a minimal montage of one EEG channel, one EOG channel, and chin EMG. Concretely we use EEG at Fpz-Cz or the study specific single EEG when Fpz-Cz is unavailable, left EOG referenced to PG1, and chin EMG. Table 1 lists subject counts and native sampling.

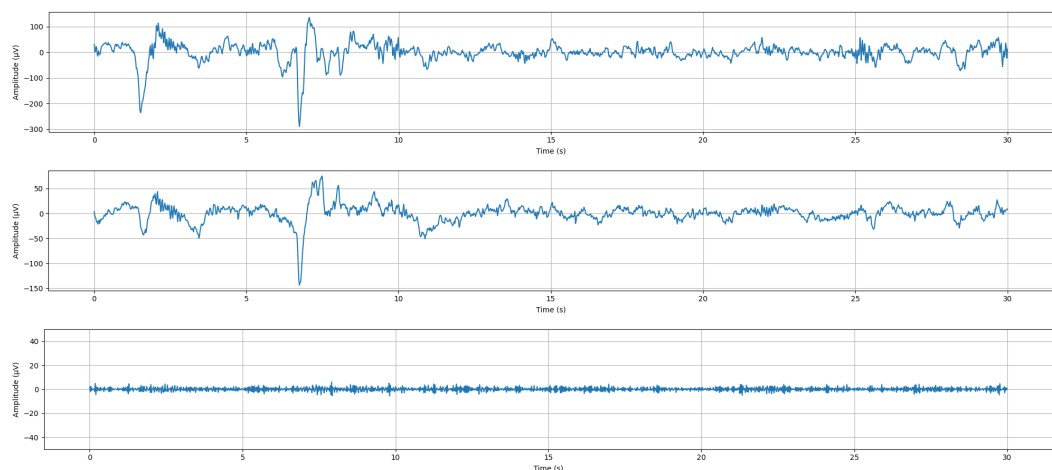

Figure 1: Example curated epoch image obtained by stacking EEG, EOG, and EMG from top to bottom. The example shows stage N2 with characteristic morphology.

**DCSM** Perslev et al. (2021) The Danish Center for Sleep Medicine cohort provides open access overnight polysomnography from a healthy population collected between two thousand fifteen and two thousand eighteen. Expert scorers followed AASM rules. The native montage includes multiple frontal central and occipital EEG derivations with bilateral EOG at a sampling rate of two hundred and fifty six hertz. For our benchmark we harmonize to a minimal montage that matches the image stack protocol. Concretely we use EEG at F4-M1, EOG at E1-M2, and chin EMG. The curated subset characteristics appear in Table 1.

**ISRUC** Khalighi et al. (2016) The ISRUC Sleep dataset was collected at the Sleep Medicine Centre of the Hospital of Coimbra University and contains overnight recordings of about eight hours with nineteen physiological channels sampled at two hundred hertz. Two expert scorers provided AASM labels. We use the labels from expert one. To align with our pipeline we adopt a minimal montage with EEG at C4-M1, EOG at E1-M2, and chin EMG. Table 1 summarizes the curated subset.

**Signal to image details.** All channels are resampled to one hundred hertz after power line notch removal and modality specific band pass filtering. We use a zero phase fourth order IIR band pass per modality as follows. EEG uses zero point three to thirty five hertz. EOG uses zero point one to ten hertz. EMG uses ten to forty five hertz. Before resampling we apply an anti alias low pass at forty eight hertz. Signals are not normalized across recordings, consistent with clinical reading practice, so that absolute amplitudes and waveform morphology remain visible without rescaling. Raw time series aligned to scorer time stamps are split into contiguous thirty second epochs, and unscored segments and movement time are excluded. For each retained epoch we construct a single vertically stacked canvas with EEG on top, EOG in the middle, and EMG at the bottom so that the three modalities share the same horizontal axis from zero to thirty seconds and follow the physician reading order. The waveform panels display amplitude in microvolts on the vertical axis and time in seconds on the horizontal axis with grid lines every five seconds. To mitigate the strong class imbalance present in full night recordings, we adopt class balanced sampling when constructing the curated image corpus and for training mini batches, targeting approximately twenty percent per stage as summarized in Table. 2. An example epoch image is shown in Figure. 4.

**Sequence model input preparation.** For sequence oriented large language models we segment the raw PSG into non overlapping thirty second epochs and use the EEG channel only. The EEG signal is downsampled to thirty hertz to control sequence length and memory cost while retaining low frequency rhythms that are informative for sleep staging. For end to end deep learning baselines we follow the same epoching strategy and resample signals to one hundred hertz, which matches the settings used in our image construction pipeline. Each sequence sample corresponds one to one with its image counterpart because both are derived from the same thirty second epoch boundaries with the same time stamps and the same stage label. This alignment ensures that comparisons across sequence inputs and image inputs are fair and that per epoch analyses and ablations can be matched exactly across modalities.

| Dataset | W | N1 | N2 | N3 | REM | Total |
|---|---|---|---|---|---|---|
| DCSM | 453 | 585 | 606 | 486 | 572 | 2702 |
| | 16.77% | 21.65% | 22.43% | 17.99% | 21.17% | 100% |
| SHHS | 327 | 494 | 605 | 472 | 492 | 2390 |
| | 13.68% | 20.67% | 25.31% | 19.75% | 20.59% | 100% |
| ISRUC | 460 | 443 | 499 | 447 | 400 | 2249 |
| | 20.45% | 19.70% | 22.19% | 19.88% | 17.79% | 100% |

Table 2: Sleep stage distribution across datasets, reported in both counts and percentages.

## 4 PHYSICIAN CONSENSUS SLEEP STAGING STANDARDS WITH EXAMPLES

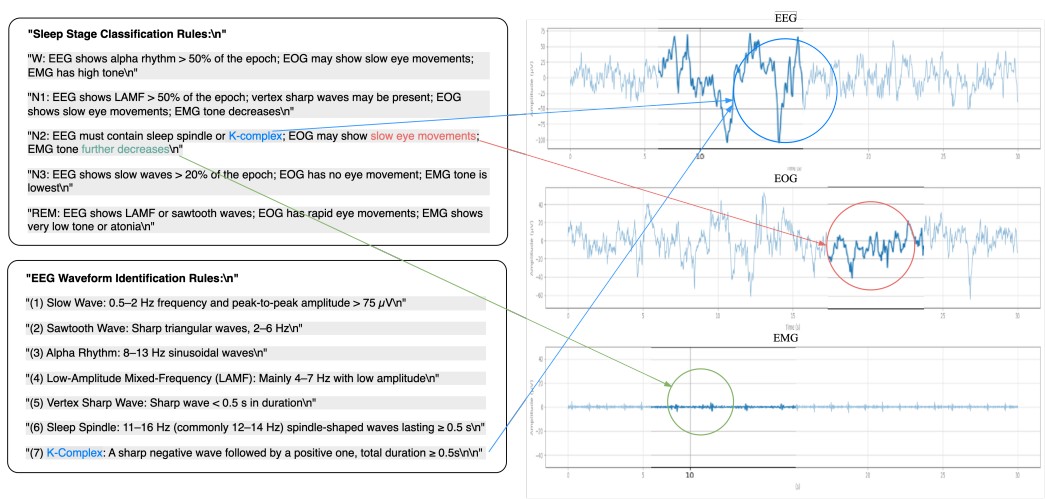

Figure 2: Example curated epoch image obtained by stacking EEG, EOG, and EMG from top to bottom. The example shows stage N2 with characteristic morphology.

Figure. 2 presents the staging criteria that we co developed with board certified sleep physicians. The criteria are grounded in the AASM guideline Berry et al. (2017) and are rewritten to make them easier for large models to follow while preserving essential information. The left panel summarizes the key determinants for each stage. Staging relies primarily on EEG, EOG confirms eye movement patterns, and EMG evaluates muscle tone and its trend, with muscle tone most strongly associated with the REM stage. Muscle tone is extremely low during REM and often shows nearly no visible waveform. Because REM and N1 exhibit similar morphology on EEG, relying on EEG alone can lead to confusion. Adding EMG markedly improves the accuracy of REM classification. The right panel shows a strictly matched thirty second stacked canvas where representative events on EEG, EOG, and EMG are marked on a common time axis, allowing readers to locate the waveform evidence that corresponds to each textual rule.

Methodologically, these clinician readable rules are used both for data curation and for model design. Physicians use them during quality control to verify waveforms and stage labels when selecting clear exemplars, and the same rules are distilled into concise expert feature phrases that are injected as prompts for large language models so that the models can reason with explicit physiological cues. Because the textual rules and the thirty second images are paired one to one, the model's stage predictions and rationales can be traced back to concrete waveform evidence, which facilitates interpretability and the review of error cases.

As an example for N2, the rule requires the presence of a sleep spindle or a K-complex in the EEG Berry et al. (2017). In the figure, the EEG trace highlights the typical K-complex pattern,

namely a sharp negative deflection followed by a positive deflection with a total duration of at least zero point five seconds. When a spindle is present, it lies within eleven to sixteen hertz, has a spindle shaped envelope, and lasts at least zero point five seconds. Consistently, the middle EOG trace shows slow eye movements, and the lower EMG trace shows further reduction of muscle tone. These three pieces of evidence within the same thirty second window correspond directly to the three rules that the EEG must contain a spindle or a K-complex, the EOG may show slow eye movements, and the EMG shows a further decrease in muscle tone, making the staging basis verifiable on the image.

# 5 EXPERIMENTS

## 5.1 EXPERIMENTAL RESULTS

| Dataset | Models | Images only | | | With stage features | | |
|---------|--------|------|------|------|------|------|------|
| | | acc | f1 | rec | acc | f1 | rec |
| **DCSM** | GPT-4o | 21.35 | 16.55 | 20.07 | 20.43 | 16.64 | 19.04 |
| | Gemma-3-27b-it | 19.89 | 13.69 | 19.39 | 21.82 | 18.55 | 21.65 |
| | Qwen2.5VL-7b | 20.95 | 10.89 | 19.92 | 28.13 | 23.61 | 28.00 |
| | QwenQVQ-72b | 26.22 | 23.34 | 23.84 | 26.21 | 22.01 | 23.75 |
| | Llama-3.2-11B | 20.76 | 17.31 | 19.18 | 22.24 | 20.60 | 21.46 |
| | Llava-1.5-7b | 25.35 | 19.47 | 23.53 | 20.80 | 11.62 | 18.69 |
| | **AVG** | **22.42** | **16.88** | **20.99** | **23.27** | **18.84** | **22.10** |
| **SHHS** | GPT-4o | 22.34 | 17.50 | 20.71 | 23.43 | 19.01 | 21.81 |
| | Gemma-3-27b-it | 26.23 | 19.02 | 21.93 | 26.98 | 23.66 | 22.76 |
| | Qwen2.5VL-7b | 20.50 | 10.81 | 19.89 | 26.95 | 22.80 | 26.48 |
| | QwenQVQ-72b | 25.33 | 22.00 | 22.87 | 22.45 | 17.86 | 20.18 |
| | Llama-3.2-11B | 21.80 | 17.99 | 18.94 | 21.42 | 19.98 | 20.14 |
| | Llava-1.5-7b | 25.06 | 19.84 | 23.52 | 23.64 | 11.73 | 18.94 |
| | **AVG** | **23.54** | **17.86** | **21.31** | **24.15** | **19.17** | **21.72** |
| **ISRUC** | GPT-4o | 19.66 | 15.13 | 20.54 | 19.97 | 14.79 | 20.80 |
| | Gemma-3-27b-it | 19.87 | 12.43 | 16.25 | 20.97 | 17.88 | 20.65 |
| | Qwen2.5VL-7b | 16.15 | 6.92 | 17.97 | 21.98 | 19.19 | 22.43 |
| | QwenQVQ-72b | 25.61 | 21.96 | 25.59 | 23.41 | 20.37 | 21.64 |
| | Llama-3.2-11B | 21.93 | 17.19 | 21.11 | 19.22 | 17.38 | 19.84 |
| | Llava-1.5-7b | 20.42 | 16.31 | 20.64 | 22.38 | 11.69 | 20.53 |
| | **AVG** | **20.61** | **14.99** | **20.35** | **21.32** | **16.88** | **20.98** |

Table 3: VLM results on three cohorts using EEG plus EOG plus EMG. For each method we report accuracy, macro F1, and recall under two settings images only and with expert stage features. Values are percentages.

We evaluate a representative set of recent large vision–language models spanning both proprietary and open-source families. Specifically, our benchmark includes: GPT-4o Hurst et al. (2024), a frontier proprietary multimodal model widely regarded as a strong general-purpose baseline; Gemma-3-27b-it Team et al. (2025), an instruction-tuned model optimized for vision-language reasoning; Qwen2.5VL-7b Bai et al. (2025), a compact yet competitive open-source VLM; QwenQVQ-72b Team (2024), a very large open-source model with enhanced multimodal pretraining; Llama-3.2-11B Meta (2024), which represents the Llama family extended to multimodal inputs; and Llava-1.5-7b Liu et al. (2024a), a widely used open-source multimodal model. Table. 3 summarizes large language model results on three cohorts using EEG, EOG, and EMG, comparing two input settings: images only versus images plus expert stage features. Overall performance is low: averaged across datasets, accuracy is about 20–24%, macro F1 about 15–19%, and recall around 21%. This contrasts sharply with prior end to end deep models on the same data and indicates that current VLMs are not yet adequate for standalone, clinical grade staging.

Adding expert stage features usually yields modest gains, especially on DCSM and SHHS, where most models improve in accuracy and recall. For example, on DCSM Qwen2.5VL raises accuracy from 20.95% to 28.13% and macro F1 from 10.89% to 23.61%. Cohort means also increase, as

visualized in Figure. 3: on DCSM the average accuracy rises from 22.42% to 23.27% by roughly 0.8 percentage points, and on SHHS from 23.54% to 24.15% by about 0.6 percentage points. However, these gains are far from closing the gap to strong task specific baselines, and some models even degrade when features are provided; for instance, Llava shows a clear drop in macro F1 across datasets, reflecting heterogeneous sensitivity of multimodal front ends to textualized features.

Cross cohort differences are also evident. ISRUC yields the lowest overall numbers in both accuracy and macro F1, suggesting that VLMs generalize poorly in cohorts with heavier pathology or greater population heterogeneity. SHHS attains slightly higher cohort means than DCSM when features are included, but remains in a low performance range. By model class, the larger QwenQVQ-72b and the feature augmented Qwen2.5VL-7b are relatively more stable, whereas GPT-4o shows no clear advantage. These findings reinforce our central conclusion that even a state of the art multimodal model cannot surpass strong baselines under our fully supervised and cross dataset settings.

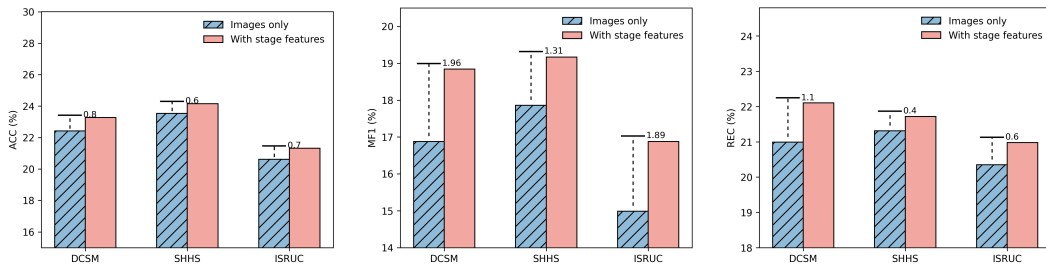

Figure 3: Average gains from adding expert stage features across DCSM SHHS and ISRUC in ACC MF1 and REC.

| Dataset | Model | acc | f1 | rec |
|---------|-------|-----|-----|-----|
| **DCSM** | ChatTS-14B | 21.63 | 17.19 | 20.60 |
| | DeepSleep | 65.23 | 66.73 | 66.52 |
| | AttnSleep | 69.34 | 70.52 | 71.22 |
| **SHHS** | ChatTS-14B | 18.95 | 18.33 | 20.53 |
| | DeepSleep | 59.01 | 59.66 | 60.57 |
| | AttnSleep | 63.64 | 64.91 | 64.33 |
| **ISRUC** | ChatTS-14B | 21.52 | 21.02 | 21.41 |
| | DeepSleep | 48.06 | 48.46 | 47.67 |
| | AttnSleep | 56.03 | 56.54 | 56.09 |

Table 4: Results with raw sequence inputs. We report accuracy, macro F1, and recall for each model on each cohort. Values are percentages.

The results in Table.4 compare sequence based methods using raw inputs from a single EEG channel. Conventional deep learning baselines clearly dominate. On DCSM, AttnSleep attains 69.34 percent accuracy, 70.52 percent macro F1, and 71.22 percent recall, while DeepSleep reaches 65.23, 66.73, and 66.52 percent. By contrast, ChatTS-14B Xie et al. (2024) records 21.63 percent accuracy, 17.19 percent macro F1, and 20.60 percent recall, which is close to chance for a five class task. The same pattern holds on SHHS and ISRUC, where AttnSleep and DeepSleep exceed ChatTS-14B by roughly thirty to fifty percentage points across metrics. The large gap between sequence large language models and task specific deep networks further supports our conclusion. Even under the favorable single channel setting, the sequence understanding of a general purpose language model is not a substitute for architectures with inductive bias for physiological time series. Likely causes include limited representation of long duration sleep rhythms and the absence of constraints that encode frequency band structure and waveform morphology. Taken together with the image and feature results, these sequence experiments reinforce that current large language models remain insufficient for independent clinical deployment in sleep staging.

## 6 Discussion

| Prompt | Prompt |
|---|---|
| "This image shows EEG, EOG, and EMG signals from a 30-second sleep epoch (from top to bottom). Please determine which sleep stage (W, N1, N2, N3, REM) this period is most likely to belong to, and briefly explain your reasoning" | "This image shows EEG, EOG, and EMG signals from a 30-second sleep epoch (from top to bottom). Determine the most likely sleep stage (W, N1, N2, N3, REM) and explain why. Use the following EEG waveform and sleep stage classification criteria:\n\n""Sleep Stage Classification Rules:\n"".....""EEG Waveform Identification Rules:\n"".....".." |
| GPT-4o Answer | GPT-4o Answer |
| "tp169bpsg_1529_N1.png": {
"result": "Based on the provided signals, this epoch is most likely in the N2 sleep stage (light sleep). Here's the reasoning for this:\n\n1.
**EEG (Top panel):** \n  - The EEG signal exhibits a mixed frequency pattern with some low-amplitude activity. While there are some fluctuations, the absence of consistent high-frequency activity rules out Wakefulness (W) and Rapid Eye Movement (REM) sleep. Additionally, the lack of distinct large slow waves rules out N3 (slow-wave sleep).\n  - While sleep spindles are not distinctly visible in the image, the mid-range frequency pattern can imply their presence, which is characteristic of N2.\n\n2.
**EOG (Middle panel):** \n  - The EOG shows relatively low activity, with no obvious rapid eye movements present, which suggests that this is not REM sleep. NREM stages, including N2, commonly show low EOG activity, except for some slow rolling eye movements in N1, which are not clearly observed here.\n\n3.
**EMG (Bottom panel):** \n  - The EMG shows low levels of muscle activity, which is common in NREM sleep stages. The EMG activity here does not indicate the complete muscle atonia associated with REM sleep.\n\n Overall, the combination of EEG features suggesting possible sleep spindles, low EOG activity, and reduced EMG tone is consistent with N2 sleep."  } | "tp169bpsg_1529_N1.png": {
"result": "Based on the provided criteria and analysis of the plotted signals, the most likely sleep stage for this 30-second epoch is N1 (Non-REM Stage 1). Here's why:\n\n1.
**EEG (Top Panel)**: The EEG trace does not seem to show a consistent presence of alpha rhythm or sleep spindles. Instead, it exhibits low-amplitude mixed-frequency (LAMF) activity, characteristic of Stage N1. There's no clear indication of slow waves that are more than 20% of the epoch, which rules out stages N2 and N3. The absence of alpha rhythm (8-13 Hz) ruling above 50% of the epoch suggests it's also not the Wake stage.\n\n2.
**EOG (Middle Panel)**: The EOG trace does not clearly display rapid eye movements but may indicate the presence of slow eye movements, which are consistent with Stage N1. Slow eye movements are often associated with the transition from wakefulness to sleep.\n\n3.
**EMG (Bottom Panel)**: The EMG trace indicates some muscle tone, although it appears less than what would be characterized as high tone during wakefulness, supporting a transition to sleep. It is not as low as it would be during REM sleep atonia.\n\n Overall, the combination of LAMF EEG activity, potential slow eye movements, and decreased EMG tone fits well with the characteristics of sleep stage N1."} |
| Llava-1.5-7b Answer | Llava-1.5-7b Answer |
| "N2_tp0ae9psg_2225": "USER: \nThis image shows EEG, EOG, and EMG signals from a 30-second sleep epoch (from top to bottom). Please determine which sleep stage (W, N1, N2, N3, REM) this epoch most likely belongs to, and explain your reasoning. ASSISTANT: Based on the EEG, EOG, and EMG signals in the image, it is difficult to determine the exact sleep stage without more information. However, we can make some general observations. The EEG signal shows a mix of delta, theta, and alpha waves, which are typical for different sleep stages. The EOG signal, which measures eye movements, can provide additional information about the sleep stage. The EMG signal, which measures muscle activity, can also provide insights into the sleep stage.\n\nTo determine the sleep stage, one would need to analyze the wave patterns, frequency, and amplitude of the signals, as well as the eye movements and muscle activity. This would require a more detailed examination of the data and a thorough understanding of the sleep stages and their corresponding patterns.", | "N2_tp0ae9psg_2225": "USER: \nThis image shows EEG, EOG, and EMG signals from a 30-second sleep epoch (from top to bottom). Determine the most likely sleep stage (W, N1, N2, N3, REM) and explain why. \n\nUse the following EEG waveform and sleep stage classification criteria:\n\nSleep Stage Classification Rules:\nW: EEG shows alpha rhythm > 50% of the epoch; EOG may show slow eye movements; EMG has high tone\nN1: EEG shows LAMF > 50% of the epoch; vertex sharp waves may be present; EOG shows slow eye movements; EMG tone decreases\nN2: EEG must contain sleep spindle or K-complex; EOG may show slow eye movements; EMG tone further decreases\nN3: EEG shows slow waves > 20% of the epoch; EOG has no eye movement; EMG tone is lowest\nREM: EEG shows LAMF or sawtooth waves; EOG has rapid eye movements; EMG shows very low tone or atonia\nEEG Waveform Identification Rules:\n(1) Slow Wave: 0.5–2 Hz frequency and peak-to-peak amplitude > 75 µV\n(2) Sawtooth Wave: Sharp triangular waves, 2–6 Hz\n(3) Alpha Rhythm: 8–13 Hz sinusoidal waves\n(4) Low-Amplitude Mixed-Frequency (LAMF): Mainly 4–7 Hz with low amplitude\n(5) Vertex Sharp Wave: Sharp wave < 0.5 s in duration\n(6) Sleep Spindle: 11–16 Hz (commonly 12–14 Hz) spindle-shaped waves lasting ≥ 0.5 s\n(7) K-Complex: A sharp negative wave followed by a positive one, total duration ≥ 0.5 s\n\n ASSISTANT: Based on the EEG waveform and sleep stage classification rules, the most likely sleep stage in this 30-second sleep epoch is N2. The EEG shows LAMF > 50% of the epoch, which is a characteristic feature of N2 sleep stage. Additionally, the EOG may show slow eye movements, and the EMG tone decreases, further supporting the N2 sleep stage classification.", |

Figure 4: Example curated epoch image obtained by stacking EEG, EOG, and EMG from top to bottom. The example shows stage N2 with characteristic morphology.

For GPT-4o, the answer without rule prompts labels the epoch as N2 and cites low amplitude mixed frequency activity in the EEG together with low EOG activity and reduced EMG tone. These observations do not directly address the decisive N2 criterion, namely the presence of a sleep spindle or a K-complex, which the model neither detects nor explicitly claims to observe. After we supply the staging and waveform rules, GPT-4o changes its decision to N1 and structures the rationale channel by channel following the prompt. The argument becomes more orderly but remains internally inconsistent it states that there is insufficient evidence for wakefulness or N2 and N3, yet the support for N1 remains a generic appeal to LAMF, and the descriptions of EOG and EMG evidence are vague. The prompts therefore help organize the narrative but do not reliably improve recognition of the key events. For LLaVA-1.5-7B, the answer without rule prompts is largely evasive. It stresses that more information is needed and offers only broad observations, reflecting uncertainty and the lack of stable criteria for this task. Given the same image with rule prompts, the model restates the staging and waveform rules and outputs N2. However, the rationale does not match the rules it relies on LAMF and slow eye movements while failing to identify the mandatory N2 evidence of a spindle or a K-complex. The mention of low EMG tone is also insufficient to separate N2 from N1 or REM. Across the two examples we observe recurring failure modes. The models have weak key event detection when spindles or K-complexes are subtle, they substitute soft cues such as LAMF, slow eye movements, and low muscle tone, which invites N1 versus N2 confusion. They are prone to prompt anchoring with rules provided, the reasoning follows the prompted template even when the corresponding waveform is not detected, producing explanation like text without matching evidence. Finally, evidence integration across channels is limited. Most answers list EEG, EOG, and EMG independently rather than enforcing cross channel constraints for instance, when EEG mor-

phology makes REM and N1 look alike, the decisive clue is EMG atonia to rule out N1, yet this interaction is rarely reflected in the final decision.

# 7 CONCLUSION

This study introduces SleepVLM-Bench, a unified benchmark and evaluation protocol that couples long duration polysomnography with VLMs for clinical sleep staging. We reformulate whole-night multi-channel recordings into three VLM consumable input forms and evaluate mainstream models against strong end-to-end baselines on DCSM, SHHS, and ISRUC. Both quantitative and qualitative evidence converge on a clear conclusion: current VLMs, including GPT-4o, underperform task-specific deep neural network solutions, and their performance is insufficient for independent clinical-grade deployment. We found that injecting expert stage features improves interpretability and yields modest gains, particularly on DCSM and SHHS, yet the overall gap to strong baselines remains.

# 8 FUTURE WORK

Our results suggest several impactful directions, including event-aware perception, where large language models are combined with explicit detectors for spindles, K-complexes, slow waves, rapid eye movements, and muscle atonia, fusing symbolic event streams with time series features before inference; physiological priors and constraints, by encoding stage transition rules, stage-specific duration statistics, and cross-channel dependencies to ensure that EEG, EOG, and EMG evidence is aggregated consistently; multimodal fusion beyond signals, through integration of PSG with clinical text such as history, medication, and technician notes that human scorers routinely use; robustness and generalization, with emphasis on cross-cohort and cross-device training protocols, label-free domain adaptation, and calibration under distribution shift, particularly for cohorts with heavier pathology burden; data efficiency, by exploring few-shot and active learning regimes where expert feedback targets rare stages and ambiguous epochs, guided by uncertainty-based sampling; efficient models, designed to run in real time with privacy-preserving deployment in clinics and sleep labs; and interpretability by design, pairing textual rationales with verifiable waveform evidence and ensuring disagreements are auditable through traceable links between predictions and supporting events.

## ETHICS STATEMENT

This study makes use of publicly accessible sleep datasets (DCSM, SHHS, ISRUC) that are widely employed in the literature. The approach presented here is intended strictly for research and methodological exploration. It is not certified for direct clinical application, and any future medical deployment would necessitate thorough regulatory review and oversight by qualified professionals.

## REPRODUCIBILITY STATEMENT

We will release the related code to reproduce our experimental results, ensuring that our findings can be replicated and extended by the community.

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

## A  LLM USAGE

Large Language Models were used to assist in refining the manuscript's language. Their role was limited to tasks such as sentence rephrasing, grammar correction, and improving clarity and readability across sections, helping to enhance the overall flow of the text. The LLMs were not involved in developing ideas, research methodology, experimental design, or data analysis. All scientific content and findings were created by the authors, who take full responsibility for the manuscript. The use of LLMs complied with ethical standards, ensuring no plagiarism, misrepresentation, or scientific misconduct.

