# OpenReview forum: "Asleep at the Wheel: Benchmarking the Inattention of Vision Language Models to Clinical Sleep Signals"
_ICLR.cc/2026/Conference — Submitted to ICLR 2026_

### Official Review · Reviewer_KKub · 2025-10-15

**Soundness:** 2
**Presentation:** 3
**Contribution:** 2
**Rating:** 4
**Confidence:** 4

**Summary:**

This paper introduces SleepVLM-Bench, a unified benchmark designed to evaluate VLMs for sleep stage analysis. The authors reconstruct full-night PSG recordings into three types of VLM-consumable inputs—time–frequency images, clinical rule-based text, and raw/discretized sequences—and evaluate multiple state-of-the-art VLMs under both subject-wise and cross-cohort settings on three datasets. The benchmark also compares these models against strong end-to-end deep learning baselines. The main conclusion is that current VLMs perform significantly worse than task-specific temporal models and are far from clinically deployable; expert rule prompts offer marginal gains in interpretability but do not close the performance gap.

**Strengths:**

1.	First clinical benchmark tailored for VLMs – Provides unified preprocessing, data splits, and metrics, ensuring reproducibility and fair comparison.

2.	Systematic multi-modal input design – Evaluates image, text, and sequence modalities in parallel, clearly illustrating the trade-offs across representation forms.

3.	Clinical rule alignment – Textual encoding of AASM criteria paired with visual evidence enhances interpretability and traceability.

4.	Clear and impactful conclusion – Offers valuable negative findings that delineate the current boundary of VLM applicability in clinical sleep analysis.

**Weaknesses:**

1. Limited methodological novelty: The paper mainly focuses on dataset construction and evaluation rather than proposing new architectures or learning algorithms. Its contribution lies more in analysis and benchmarking than in algorithmic innovation.
2. Insufficient exploitation of cross-channel information: Although multiple modalities (EEG, EOG, EMG) are included, the framework lacks explicit mechanisms for cross-channel alignment or constraint modeling, limiting its ability to leverage multimodal complementarity.
3. Potential data selection bias: The preprocessing pipeline excludes recordings with artifacts, missing channels, or labeling inconsistencies to produce “clean canvases.” While this improves input consistency, it may bias the data distribution and overestimate model performance under real clinical noise conditions.
4. Limited handling of device and montage variability: Despite including cross-dataset evaluations, the study does not address differences in sampling rates, channel montages, or recording equipment, nor propose adaptation strategies for heterogeneous data sources.
5. Lack of quantitative error analysis: While qualitative examples of VLM misclassifications are presented, there is no systematic breakdown of errors by stage or event type, making it unclear which aspects most contribute to performance degradation.
6. No discussion on computational efficiency or deployability: The paper omits practical aspects such as inference time, memory footprint, privacy concerns, or deployment feasibility in clinical environments, limiting its utility for real-world adoption.

**Questions:**

Please see Weaknesses

**Details Of Ethics Concerns:**

NAN

---

### Official Review · Reviewer_iFjG · 2025-10-30

**Soundness:** 3
**Presentation:** 3
**Contribution:** 3
**Rating:** 6
**Confidence:** 2

**Summary:**

This article introduces SleepVLM-Bench, a benchmark for evaluating Vision Language Models (VLMs) for clinical sleep staging. The authors reformatted multichannel polysomnography (PSG) data into a VLM-compatible format and rigorously tested the performance of several leading models (such as GPT-4o) in different populations.

**Strengths:**

1) The introduction of SleepVLM-Bench as an assessment framework is a significant contribution to the field, providing a standardized assessment tool for clinical sleep staging.
2) The rigorous methodological approach of screening and systematically evaluating comprehensive datasets from multiple real-world populations (DCSM, SHHS, ISRUC) ensures the fairness and robustness of model comparisons.
3) Clear research findings: The article clearly demonstrates the limitations of VLMs in clinical sleep staging, providing clear arguments that current models cannot meet clinical requirements.

**Weaknesses:**

1) While the paper clearly demonstrates that expert feature suggestions bring some improvement, the results still lag behind task-specific deep learning models. Further exploration of directions or methods for model improvement could significantly enhance the paper's impact.
2) Although evaluating existing models is valuable, proposing innovative architectures or methods to address the shortcomings of current models would likely better advance the field.
3) A more detailed discussion of the impact of VLM failures on clinical applications, along with specific improvement measures, would enhance the paper's practical significance in the field of clinical sleep.

**Questions:**

See Weaknesses for details.

---

### Official Review · Reviewer_rvtn · 2025-10-31

**Soundness:** 1
**Presentation:** 3
**Contribution:** 1
**Rating:** 2
**Confidence:** 4

**Summary:**

This paper aims to evaluate the applicability of Vision-Language Models (VLMs) for clinical sleep staging. The authors construct the SleepVLM-Bench benchmark, converting polysomnography (PSG) data into three VLM-processable modalities (time-frequency images, clinician-derived textual features, and raw/discrete sequences), and compare the performance of mainstream VLMs against specialized deep learning baselines across multiple real-world datasets. The core conclusion is that current VLMs significantly underperform specialized baselines and are unsuitable for independent clinical deployment. The primary contribution lies in proposing the first VLM-centric benchmark for this task and reporting negative results.

**Strengths:**

1. **Importance of the Problem**: Investigating the application prospects of general-purpose VLMs in critical medical domains is a topic of significant practical relevance and timeliness.

2. **Systematic Attempt**: Represents the first systematic effort to construct a VLM-focused evaluation framework for sleep staging and explores multiple data adaptation pathways.

**Weaknesses:**

#### **1. The True Potential of VLMs Remains Questionable**

1.1) **Insufficient evaluation of transformation information loss**: The authors convert raw PSG signals into images using a single, relatively simplistic method (resulting in time-domain plots, mislabeled as time-frequency images). The potential performance impact due to information loss during this conversion (e.g., loss of precise frequency components, subtle amplitude changes) is not adequately assessed. A more thorough evaluation, potentially involving ablation studies with different "time-series to image" transformation techniques, is necessary.

1.2) **Lack of VLM Adaptation**: Although the paper evaluates several recent VLMs in a zero-shot/few-shot setting, it remains unclear if their inherent visual-linguistic knowledge is fully leveraged. More comprehensive experiments, such as post-training paradigms (e.g., few-shot fine-tuning) or semantic alignment learning across modalities using limited examples, are required to truly unlock the VLM's potential and fairly judge their actual performance in this domain.

#### **2. Inadequate Baseline Selection and Insufficient Benchmark Depth**

2.1) **Non-comprehensive Baselines**: Given the goal of providing a VLM-centric benchmark, the current baseline selection is insufficient. It should include, but not be limited to: pure vision models (e.g., ViT), pure clinical text models (e.g., Med-PaLM, BioGPT), lightweight fusion methods for vision and text, and VLM variants incorporating temporal modeling (e.g., TimeVLM). In the current state, it is impossible to discern whether the performance difference stems from the VLM architecture itself or from suboptimal strategies for handling the input modalities.

2.2) **Limited Benchmark Depth**: The paper focuses solely on single-epoch classification performance. It fails to comprehensively evaluate VLMs on other critical dimensions relevant to sleep staging, such as whole-night sleep structure analysis (e.g., hypnogram continuity, stage transition rules), performance across different patient populations, or mechanistic explorations like the quality of cross-modal understanding or the consistency between model explanations and waveform evidence.

2.3) **Limited and Potentially Biased Dataset**: The dataset used for evaluation is notably small and consists of physician-selected "high-quality, canonical" excerpts (e.g., only 15 subjects from SHHS, which originally contains >5000). This may not represent the true data distribution. Furthermore, the paper does not clearly detail the specific **criteria and process for selecting this small data subset** (e.g., random selection vs. selection based on specific representativeness criteria). This raises concerns about whether the experimental results are accidental and whether they can be generalized, and also makes one wonder if the poor performance is partly due to the small dataset size or selection bias.

**Questions:**

#### **1. Comprehensiveness of Experimental Design and Attribution Analysis**

**Rationale for Baseline Comparison**: The paper compares VLMs using **image modality** input against specialized models using **raw signal modality** input. This design potentially confounds the effects of "model architecture differences" and "input modality differences" on the results. Given the three VLM-suitable modalities proposed, why were comparisons not introduced against models handling these corresponding modalities? For instance, comparisons with pure vision models (ViT, etc.), pure clinical text models (Med-PaLM, BioGPT), lightweight vision-text fusion methods, or VLM+temporal methods (TimeVLM) would greatly clarify the source of performance gaps.

#### **2. Completeness and Depth of VLM Capability Assessment**

**Missing VLM Sequence Modeling Evaluation**: The paper claims to evaluate three input modalities but substitutes the assessment of the "Raw/Discrete Feature Sequences" modality using an LLM (ChatTS-14B) instead of a VLM. Please explain the **rationale for using an LLM instead of a VLM for this evaluation**. Does this indicate a technical obstacle in feeding raw sequences to VLMs? Please provide VLM-specific results for this modality or revise the claims accordingly.

**Sufficiency of Model Potential Exploration**: The conclusions are based primarily on zero-shot/few-shot testing, with no **domain-specific fine-tuning** of open-source VLMs conducted. Fine-tuning is a critical step for unlocking model potential. Were any fine-tuning experiments performed? If fine-tuning leads to significant performance improvement, how would the paper's core thesis regarding "VLM inapplicability" be adjusted?

#### **3. Methodological Transparency and Result Reliability**

**Details on Result Generation and Statistics**: To ensure reproducibility, please elaborate on the following details:

- The specific **parsing pipeline** for converting VLM-generated textual outputs into sleep stage labels (including prompt templates and post-processing rules/algorithms).
- The specific measures taken to ensure **unbiased and consistent** statistics.

**Evidence for Benchmark Validity**: How is the effectiveness of the "image + text" data representation used in SleepVLM-Bench demonstrated? For example, was it validated that providing the same "data + expert rules" to a **text-only LLM** yields better performance than providing only the data? This would help distinguish the contribution of the "knowledge injection method" from the "VLM's multimodal capability".

---

### Official Review · Reviewer_RPez · 2025-11-01

**Soundness:** 2
**Presentation:** 2
**Contribution:** 1
**Rating:** 2
**Confidence:** 3

**Summary:**

This paper introduces SleepVLM-Bench, described as the first benchmark to evaluate the capability of vision-language models (VLMs) on clinical sleep staging tasks. The authors reformulate EEG, EOG, and EMG signals into visual and textual inputs and compare several VLMs (e.g., GPT-4o, Qwen-VL, LLaVA) with traditional deep learning models such as DeepSleepNet and AttnSleep across three datasets (DCSM, SHHS, ISRUC). The key finding is that current general-purpose VLMs perform poorly compared with specialized models, suggesting limited readiness for clinical deployment.

**Strengths:**

1. Evaluating multimodal foundation models on clinical physiological data is an emerging and important direction, and the paper raises a legitimate question about their current limitations. The inclusion of multiple datasets (albeit small) and several representative VLMs provides some breadth to the evaluation.
2. Demonstrating that current VLMs underperform in clinical settings is itself a useful message for the community, highlighting the gap between general multimodal understanding and domain-specific reasoning.

**Weaknesses:**

1. Misuse of the term “benchmark.”
The work does not satisfy the defining features of a benchmark:
   1) No standardized dataset split or evaluation protocol for future comparison.
   2) No leaderboard, evaluation server, or open-source baseline implementation.
   3) No unified scoring system or reproducibility pipeline.
In its current form, it is a one-off evaluation study, not a benchmark infrastructure.
2. Insufficient dataset scale and representativeness.
Each dataset subset used is very small (e.g., ~60–80 subjects) and only includes EEG/EOG/EMG channels, lacking the diversity of full polysomnography data. The aggressive signal filtering removes real-world noise, making the benchmark unrepresentative of actual clinical conditions.
3. Inconsistent and non-standardized methodology.
Input formats differ across models; the image conversion pipeline (stacked EEG/EOG/EMG) is ad hoc and not validated. Prompt templates for VLMs are hand-crafted and inconsistent across models. Baseline models are not re-trained under identical preprocessing, and some results are quoted from prior publications. These issues make comparisons unreliable and non-reproducible.
4. Lack of statistical rigor.
The paper reports single-score results without variance, standard deviation, or statistical significance analysis. As a result, it is unclear whether observed differences are meaningful.
5. Poor visualization and presentation.
Figures are informal and inconsistent in style (e.g., stacked signal plots instead of standardized benchmark charts). The visual presentation does not follow the clarity and structure expected for benchmark papers (no metric breakdowns, error analysis, or ranking visualization).
6. Limited innovation or contribution.
There is no methodological novelty, no unified framework for evaluation, and no contribution toward building sustainable infrastructure. The paper’s main message—that VLMs perform poorly—is descriptive but lacks analytical depth or actionable insight.

**Questions:**

1.	What elements qualify SleepVLM-Bench as a benchmark rather than a small-scale evaluation study?
2.	Are you planning to release a standardized leaderboard or evaluation interface to support reproducibility?
3.	How were baselines trained—did you re-train them using the same preprocessing pipeline, or rely on quoted numbers from prior work?
4.	How are VLM inputs (images and text prompts) standardized to ensure fair comparison across models?
5.	Could you provide statistical uncertainty (e.g., standard deviation or confidence intervals) for all reported results?
6.	Given the small data size, how do you justify the benchmark’s representativeness for clinical applications?

---

### Meta-Review · Area_Chair_xaZN · 2026-01-05

**Summary:**

The reviewers raised several serious concerns, including insufficient dataset scale and representativeness, limited innovation or contribution, the fact that the work does not satisfy the defining features of a benchmark, inadequate baseline selection and insufficient benchmark depth, problems with methodological transparency and result reliability, and a lack of quantitative error analysis. Reviewer scores are low overall, and the authors did not respond to reviewer concerns during the rebuttal phase. Based on this, I recommend rejection.

**Reviewer Concerns:**

None of the reviewer's concerns where addressed during rebuttal.

**Reviewer Scores:**

None, as authors did not attempt to address reviewer's comments.

---

### Decision · Program_Chairs · 2026-01-26

Reject